# Soil Physics Involvement in the Germination Ecology of Buried Weed Seeds

**DOI:** 10.3390/plants8010007

**Published:** 2018-12-29

**Authors:** Stefano Benvenuti, Marco Mazzoncini

**Affiliations:** Department of Agriculture, Food and Environment, Via del Borghetto, 80, 56124 Pisa, Italy; marco.mazzoncini@unipi.it

**Keywords:** soil texture, seed bank germination, soil compaction, seeding depth

## Abstract

Trials were performed to test the germination ecology of buried weed seeds as a function of physical soil conditions such as of burial depth, texture, and compaction. Indeed, these ecological conditions, due to the adopted agronomic practices, play a crucial role in modulating the seed bank germination dynamics. Experiments were carried out in open fields in confined soils (polypropylene pipes), and in the laboratory in Petri dishes. Sowing depth strongly inhibited the seed germination of the three weed species selected. This inhibition was found to be inversely proportional to the size of the soil particles. Compaction strongly increased the depth-mediated inhibition, especially in soils that were rich in clay particles, and was inversely proportional to the seed size. The physiological nature of the dormancy imposed by burial was investigated. In addition, ungerminated seeds, re-exhumed after deep-sowing for six months, were found to be in deep dormancy, especially after burial in compacted clay soil. This dormancy induction was more pronounced in weed species characterized by small seeds. Critical issues are discussed regarding weed seed bank ecophysiology and their management in sustainable agricultural cropping systems.

## 1. Introduction

One of the most important agronomic innovations in the last decades, in terms of weed control, has focused on predicting [1] and modelling [2] seedbank emergence dynamics. This information enables mechanical and/or chemical management strategies to be optimized in advance, before crop infestation. This has led to an increasing interest in studies on the relations between ecological factors that occur in the soil matrix and buried weed seed germination [3], seedling emergence [4], and seed dormancy induction or removal [5]. Temperature, humidity, or both [6], have been particularly focused on, since in the absence of particular light requirements, which are sometimes present in photosensitive species [7], they are the main factors that affect the germination dynamics of seed banks. However, the knowledge of the seedbank vertical arrangement takes on a crucial role in the germination capacity, and temperature and humidity alone are not sufficient to predict emergence dynamics in the agroecosystem. Indeed, these parameters could be sufficient to predict emergence only in the case of in natural ecosystems where the lack of soil tillage together with the poor self-burial ability of the seeds [8] concentrates the seed bank uniformly in the most superficial millimeters/centimeters of the soil, therefore resulting in their germination response to the hydro-thermal dynamics becoming more predictable.

Conversely, in agroecosystems, cultivation systems based on diversified soil tillage techniques (plowing, minimum tillage, no-tillage, etc.) involve a further ecological factor that is of crucial importance in modulating the weed field emergence dynamics, i.e., seed burial depth. In fact, the greatest constraint that limits seed bank germination is burial, which depends on an inhibition proportional to the distance from the ground’s surface. This behavior is not primarily due to hypoxia, but to the poor gas exchange in the micro-environment surrounding the buried seed [9]. This mechanism is mediated by the production of volatile and toxic [10] fermentation metabolites (acetaldehyde, methanol, and acetone), which are elicited by hypoxic conditions.

The depth-mediated germination inhibition is thus not directly attributable to pre-existing soil hypoxia but rather to the hypoxic conditions that occur when the seed respiration rate increases during preliminary germination. During this phase, soil limitations on oxygen replenishment tend to intensify the hypoxia already induced by the partial gas impermeability of the seed coat [11]. Consequently, the distance from the soil surface has a crucial role in allowing or preventing germination. Indeed, the gas diffusion towards the normoxic atmosphere, above the ground, becomes the only way to remove these toxic metabolites, which are capable not only of inhibiting germination but also of inducing secondary dormancy [12].

The ability to move away from the toxic volatile metabolites is therefore crucial in allowing or preventing buried seed germination. Soil texture thus plays a key role in the seed germination ecology [13], since the particle size is able to favor or hinder the gas diffusion towards the soil surface [14]. The greater air permeability in sandy soils limits hypoxia (higher velocity oxygen exchange with the soil surface) and removes toxic volatile metabolites. The depth-mediated inhibition is thus more marked in silty and/or clay soils, since, in these cases, the macroporosity (ground spaces occupied by air at the field capacity and therefore able to allow gas diffusion) is very low. In addition, this inhibition has been found to be inversely proportional to seed weight [4], since smaller seeds concentrate their energy reserves in fatty substances which involve a greater amount of oxygen availability during their pre-emergence heterotrophic phase.

It has been demonstrated that the oxygen requirement for germination in stressed hypoxic conditions [15] differs according to the type of starchy or fatty seed endosperm, since starchy reserves have been found to have a lower oxygen requirement than species with predominantly fatty reserves [16]. It is therefore likely that the seedbank itself is characterized by different emergence capacities by increasing the depth as a function of the seed reserve typologies of the various weed species. Note that silty and/or clay soils tend to compact [17], thus leading to a further reduction of gaseous diffusion and, consequently, resulting in a germination restriction.

Despite the growing interest in understanding and modeling the emergence dynamics of seed banks, information on the influence of compaction on the ecology of buried seeds is extremely poor. The aim of these experiments was to evaluate the germination ecology of weed seeds as a function of burial depth in diversified physical conditions such as soil texture and compaction. In the experiments, seeds were used from species with markedly diversified seed weights under the hypothesis that this feature may be related to germination inhibition due to burial depth.

## 2. Materials and Methods

### 2.1. Plant Material

In the summer of 2015, seeds from three different weed species (*Abutilon theophrasti*, *Polygonum convolvulus*, and *Portulaca oleracea*) were collected from agricultural environments near Pisa (Tuscany, Italy). These species were selected on the basis of their importance as weeds and also due to their differences in seed size and weight. The mother plants had grown in normal conditions without any symptoms of biotic or abiotic stress. The seeds were extracted from the respective fruits in the laboratory, cleaned, dried (max 12% humidity), and kept in glass containers (50% of air relative humidity) at 20 °C. The seed weight of each species was determined by the weight of 1000 seeds chosen randomly according to ISTA (International Seed Testing Association) rules for seed testing [18].

### 2.2. Emergence Test

Seedling emergence tests were conducted for each of the selected weed species in spring 2016 (April-May) in the experimental gardens of the Department of Agriculture, Food and Environment of Pisa University. During this spring period, mean temperatures ranged (min/max) around 15/25 °C. These sowing periods were chosen to simulate the common sowing time of spring-summer weeds. Three agricultural soils were chosen according to different textures: sandy, loam, and clay (Table 1). These soils were taken from three agroecosystems located in the province of Pisa. To prevent the undesired interference of the pre-existing seed bank, soils were obtained by excavations from a depth of over 0.4 m (below the plough pan) and the relative soil characteristics were similar to a normal topsoil and no particular problem based on aerobic or anaerobic conditions was found.

Emergence experiments were carried out in these confined soils. Thus, polypropylene pipes (10 cm in diameter and 30 cm in length) were buried vertically in the ground and filled with (weed seed-free) soil sifted through a 1-mm screen (hereafter ‘tilled soil’).

Two hundred seeds of each weed species were sown in polypropylene pipes (three replicates), following about eight months after collection, at the following depths: 0.5, 1, 2, 4, and 8 cm. These burial depths were derived from previous experiences with weed species [4], which are often already inhibited by a sowing depth of a few millimeters [19]. The same, above cited, soil types were compacted in order to simulate their frequent physical feature in real field conditions after agronomic (i.e., trafficability) and/or climatic (i.e., rain) events. After weed sowing at the same aforementioned depths, soil compaction was carried out. This compaction was achieved by placing a rigid aluminum disc (thickness 1 cm) with a diameter slightly smaller than the polypropylene pipes, subjecting it to a weight of 5 kg (about 160 g·cm^−2^) under conditions of water saturation (for 1 h). The experimental treatments were thus set: 3 species × 3 soil texture × 2 compaction degree × 3 replicates = 54 soil samples. Additional polypropylene pipes, with only the non-sown soil, confirmed (data not shown) the virtual absence of any seed banks. In addition, the area surrounding the polypropylene pipes was kept free of weeds to avoid any unwanted “seed rain” inside the cylinders.

During the emergence test, irrigation was unnecessary, since the soil moisture was under optimal conditions as a consequence of the natural humidity rise of a high water table. Emerged seedlings were counted (cotyledons appearance) and removed using laboratory tweezers. Experiments were stopped after five weeks when no further emergence was detected.

### 2.3. Calculation of Bulk Density

The bulk densities of the same soils used for emergence tests were calculated. This evaluation was carried out in additional (no-sown) polypropylene pipes filled with both tilled and compacted soil.

Undisturbed soil samples were collected from the upper (0–5 cm) layer (of each texture for both tilled and compacted soil) of the different soils using 40 mm (diameter) × 54 mm (height) cylindrical cores. Soil bulk density was determined from oven-dried undisturbed cores as mass per volume of oven-dried soil. Each measure was replicated three times.

### 2.4. Evaluation of Burial Inhibition

Emergence tests were carried out with half of the pipes having compacted soil and the other half not (tilled soil). The emergence inhibition due to soil compaction was calculated as a percentage of the difference between the tilled and compacted soil in each of the three different soil textures. For all species, the inhibition was calculated at the sowing depth of 2 cm. This percentage was plotted with the 1000 seed weight of each weed species.

### 2.5. Ungerminated Seed Recovery and Germination Test

After the emergence test, soil from the deeper sowing depth (8 cm) of the polypropylene pipes (containing ungerminated seeds) was collected, washed, and sieved to exhume seeds. In order to obtain a sufficient quantity of seeds, additional pipes were sown at this deep burial depth. This deep burial was kept for six months (October 2016). A fine metal sieve (400 µm) was used for seed recovery. Ungerminated seeds, retrieved from each polypropylene pipe (of each soil type and compaction degree) were placed in Petri dishes (50 seeds each) equipped with filter paper (Whatmann No. 1) moistened with distilled water, and incubated in climatic cabinets with a photoperiod of 12/12 h (alternating dark-light). The incubation temperature was 20/30 °C according to the temperature requirements for the germination of this species derived from previous experience. A light source of about 200 μmol·m^−2^·s^−1^ was obtained using fluorescent tubes (PHILIPS THL 20W/33, Amsterdam, The Netherlands). The germinated seed count (cotyledons appearance) was completed after four weeks when germination had essentially stopped. Ungerminated seeds (already dormant in spite of their incubation) underwent chilling (4 °C in the dark, on moistened filter paper for one month) to induce dormancy-breaking. They were then re-incubated under the same conditions. Seeds that germinated following this treatment were classified as light-dormant. Seeds remaining dormant despite this treatment were subjected to a seed-crushing test [20] to distinguish viable (deep-dormant) from dead seeds. Four replicates of 50 seeds each were used for each soil texture in both tilled and compacted conditions.

### 2.6. Statistical Analyses

For the seedling emergence and seed germination experiments, the experimental design was a factorial block with three replications. After a homogeneity test of variance, arc-sin transformation was necessary for percentage values. Data were subjected to analysis of variance (ANOVA) using the Student–Newman–Keuls test (*p* < 0.05 and *p* < 0.01) for mean separations (least-significant difference, LSD). For each statistical analysis, commercial software (CoStat, CoHort Software, Minneapolis, MN, USA) was used.

## 3. Results

Figure 1 shows the bulk density of the three different soil types (sandy, loam, and clay) before and after compaction. The compaction degree was inversely proportional to the soil particle size. In fact, the sandy soil showed only a slight compaction (about 2%), while much higher values were shown by the loam soil (about 6%) and, overall, by the clay soil (about 14%). For each of these soil physical conditions, the germination and emergence performances of the three different weed species were tested.

Figure 2 shows the percentage of seedling emergence from each of the five seeding depths (0, 2, 4, 6, and 8 cm). As can be observed, the emergence decreases at increasing seeding depths. However, this does not occur in a uniform way but is strongly dependent on both the soil texture and soil compaction. Overall, the major differences between the soft (tilled) and compacted soil (often statistically significant for *p* < 0.05 and/or *p* < 0.01) were observed at intermediate seeding depths (2–4 cm). In addition, the double inhibition, due to the sowing depth and soil compaction, was different in the three weed species. In fact, while *A. theophrasti* maintained an, albeit scarce (roughly 20%), seedling emergence even when buried in clay soil at 4 cm, *P. convolvulus* and *P. oleracea*, a strong seedling emergence inhibition was shown (emergence of only 7% and 3%, respectively). This soil depth-inhibition, above all in the clay soil, was further amplified by compaction. Indeed, the high bulk density of the compacted clay soil completely inhibited the seedling emergence of *P. convolvulus* and *P. oleracea* already at a burial of 4 cm. Moreover, their emergence in the compacted clay soil was greatly inhibited by only two centimeters of burial depth (emergence of 15% and 5%, respectively). It was hypothesized that such a diversified inhibition performance could depend on the seed weight. These values were thus measured for each weed species. Table 2 shows a huge difference in their 1000 seed weight.

In fact, while the rather large and heavy seeds of *A. theophrasti* showed (Figure 3) a poor inhibition due to compaction (under the light burial of 2 cm), those of *P. convolvulus*, and particularly *P. oleracea*, showed a marked inhibition due to soil compaction, especially in the clay texture (statistically significant for *p* < 0.05). The germination inhibition due to the compaction of clay soil was almost 70% in the case of *P. oleracea*. Similarly, a consistent but decidedly lower inhibition (almost 40%) was shown by *P. convolvulus*. Conversely, the heaviest seeds (*A. theophrasti*) showed a much lower inhibition (about 25%). Finally, when compacted, the sandy soil showed a lower ability to inhibit the seed germination, while the loam soil showed an almost intermediate behavior.

The inhibition of the compacted soil was thus calculated as a percentage with respect to the tilled soil at a sowing depth of 2 cm (for each soil texture and for each of the three species). Figure 3 highlights that this soil compaction-mediated inhibition was strongly influenced by the weight of 1000 seeds. This inhibition was found to be inversely related to the seed weight. In fact, while the heavy seeds of *A. theophrasti* showed a poor dependence on the soil texture, conversely, the much smaller *P. oleracea* seeds showed a strong inhibition (statistically significant for *p* < 0.05), especially in the case of the clay soil (almost 70% inhibition to respect to tilled soil). The inhibition showed by *P. convolvulus* was almost intermediate among the others (almost 40% in the case of the clay soil) according to the relatively intermediate 1000 seed weight.

Table 3 shows the germination performances of the three species after burial. All species showed a marked and statistically significant (*p* < 0.01) effect due to soil compaction in the clay soil in terms of decreased germination and increase in deep dormancy. No significant differences were observed in terms of light dormancy and seed vitality.

On the other hand, the sandy soil compaction showed no significant effects in terms of germination or dormancy regardless of the weed species. The loam soil showed intermediate performances which were weed-species dependent. In fact, after seed burial in this soil texture, the deep dormancy degree showed no appreciable increase in the case of *A. theophrasti*, while a consistent and significant (*p* < 0.05) increase was found in *P. convolvulus*, and even more so (*p* < 0.01) in *P. oleracea*. In general, the small seeds of *P. oleracea* showed the greatest susceptibility to the dormancy induction (light + deep) following clay soil compaction. Figure 4 shows this increased dormancy in *P. oleracea* both before and after burial in tilled clay soil (30% and 37%, respectively) and its strong dormancy induction (85%) under clay soil compaction.

## 4. Discussion

As expected, the three selected soil types, with different textures, showed not only a different bulk density but also, above all, a different susceptibility to compaction (Figure 1), which is in line with previous experiments [21]. It is well known that the weed seed bank is strongly influenced by these physical soil characteristics in terms of both seeding depth [4] and texture [13], and the present experiments fully confirm that this behavior also occurs for these species. The most important new information is related to the germination response to compaction of buried seeds in different types of soil. Indeed, the hypothesis that the soil compaction degree strongly hinders germination, and that this phenomenon is proportional to the clay content, was fully demonstrated by the data obtained (Figure 2). In fact, while the sandy soil showed a modest inhibition for weed seed germination and seedling emergence, conversely, the loam soil, and even more so the clay soil, showed a strong inhibition following compaction.

This inhibition due to compaction is particularly evident at intermediate depths (about 2 and 4 cm) in the clay soil. This is due to the strong tendency to compaction of soils rich in clay particles [22] because they are very susceptible to a reduced porosity and, consequently, gas diffusion [23]. It is therefore evident that soil gas diffusion is a very limiting factor which strongly affects weed seed bank germination, although how this occurs still needs explaining. The buried seed, in the absence of thermal and water limitations, activates germination, which involves a sudden respiration increase [24], with a consequent increase in oxygen requirement [25]. The seed coat itself acts as an obstacle to the availability of oxygen for the embryo [11].

The double action of the poor soil porosity and poor seed coat oxygen permeability creates hypoxic conditions in the soil matrix surrounding the buried seed. Such hypoxia elicits fermentative volatile metabolites such as acetaldehyde, methanol, and acetone [10], which play a well-known toxic role in seed germination [26]. As a result, soil gaseous diffusion is key to allowing their removal, and consequently, in enabling or preventing germination. This strong gas diffusion requirement, allowing the oxygen entry from the overlying atmosphere (above the ground surface) and removing the toxic volatile metabolites, explains the extreme germination inhibition generated by a compacted clay soil. However, the tolerance to the restriction of soil gas diffusion is not the same in the various weed species. Indeed, their size and weight revealed a different soil-depth inhibition. Larger and heavier seeds, as with *A. theophrasti*, have been shown to be able to perform germination and emergence in greater sowing depths [27]. Similarly, an inverse relationship between seed weight and depth-mediated inhibition has been found in several weed species [4].

A further hypothesis was thus formulated and tested: seed size may be involved not only in the ability to germinate and emerge at high sowing depths but also leads to greater tolerance to soil compaction. In order to perform a comparative evaluation of the compaction-tolerance of the three weed species, the percentage difference between tilled and compacted soils was calculated. The inhibition at a moderate burial depth (2 cm) was selected, since at this sowing depth, all species showed germination in each soil texture and compaction. These values were put into relation with the respective weight of 1000 seeds. The small and light *P. oleracea* seeds showed the greatest inhibition due to the compaction in soils rich in clay particles (loam and particularly clay soils), which could be linked to the seed endosperm reserve (starchy of fatty). Indeed, crop species characterized by starchy reserves have a lower oxygen requirement than species with predominantly fatty reserves [14].

It is therefore likely that buried weed seeds are characterized by differences in soil depth emergence capacity in relation to the type of seed reserve. In fact, these species have been found to be rich in fatty acid substances [28], unlike the soil compaction-tolerant *P. convolvulus* [3]. In addition, although it is not previously established literature the type of *A. theophrasti* reserve substances has been found that this species, characterized by relatively large and heavy seeds, sometimes is susceptible to fatal germination [29]. This germination, which is also called suicide, occurs when a weed seed germinates but the seedling dies before reaching the soil surface. In practice, this appears to confirm the lower soil gas diffusion requirements of this species to trigger germination. On the other hand, the major energy reserves of this species imply a less crucial perception of the soil surface, in terms of survival chances, due to their higher heterotrophic ability during pre-emergence growth. Conversely, very small seeds are unlikely to germinate when buried too deep in clay and compact soils. However, the small seeds that are unable to reach the soil surface, due to excessive burial and compaction, are able to perceive this risk in advance and enter a physiological phase of dormancy.

This happens mostly in compacted clay soils. Indeed, a strong dormancy induction was shown in *P. convolvulus* (Table 3) and particularly in *P. oleracea* (Figure 4) seeds. The physiological reason for this behavior appears to be closely linked to the difficulties in removing the volatile metabolites triggered by hypoxia. Indeed, the sandy soil, which is typically rich in macroporosity even when compacted [30], did not show this dormancy induction irrespectively of the weed species and their relative seed weight. On the other hand, sandy soil leads to a lower inhibition of the buried seed [13], allowing regular germination and emergence even at greater depths.

Essentially, secondary dormancy (cyclically induced and removed, [31]) tends to prevent germination from completing its biological cycle during excessive temperatures and/or burial, drought, flooding, etc. In other words, soil compaction tends to induce a hypoxia-mediated deep dormancy [32] as a survival strategy in order to overcome adverse ecological burial conditions.

In summary, soil compaction tends to elicit a persistent seed bank (via secondary dormancy induction), especially in the case of soil clay particles. This mechanism appears to complicate the defense strategies of crop rotation, since this weed seed dormancy favors seed bank longevity and, consequently, the periodic reappearance of species that had apparently disappeared from the agroecosystem.

## 5. Conclusions

The need for plant environmental adaptation [33], including weeds [34], has led to the evolution of several mechanisms that enable species to perceive which of their surrounding ecological conditions would allow for their survival. The ability of the seed bank to perceive the surrounding micro-environment thus plays a key role in the choice between germination trigger or dormancy induction. This perception prevents, or minimizes, the suicide germination [35] of seeds that are too far from the soil surface and consequently have limited energy potential during pre-emergence heterotrophic elongation. The detection capacity of soil compaction, especially if clayey, prevents a probable failure to survive of most of the seed bank, above all the small seeds. The dormancy induction makes it possible to delay germination in relation to possible favorable future events (soil tillage, soil structure dynamics, etc.) in order to synchronize the germination in the most suitable periods.

In summary, clay soil compaction favors the accumulation of a persistent seed bank. This complicates weed control strategies since the emergence dynamics [36], which are of crucial importance in terms of agronomic management, are more difficult to model and predict. Future experiments are thus necessary to improve the knowledge of the relationships between the ecological conditions of the seed bank and their germination/dormancy dynamics, thus optimizing the agronomic management in sustainable cropping systems.

## Figures and Tables

**Figure 1 plants-08-00007-f001:**
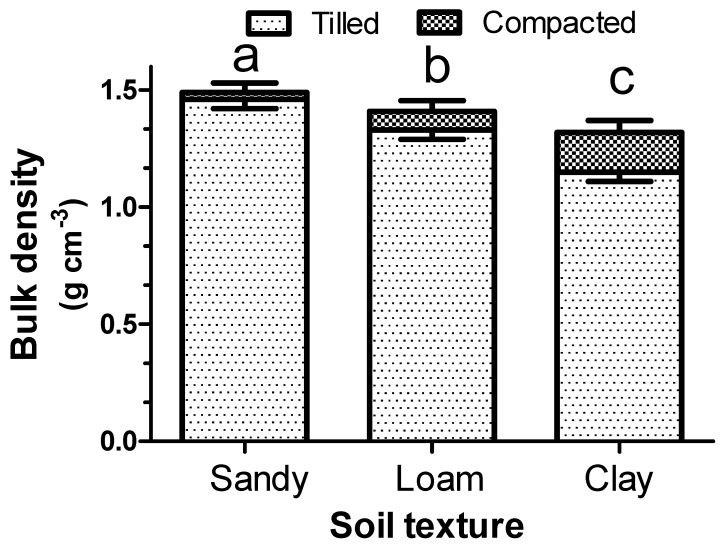
Bulk density (g·cm^−3^) of the three different soil textures (sandy, loam, and clay) before (tilled) and after soil compaction. Different letters indicate the significance of the bulk density change (before/after soil compaction) to the Student–Newman–Keuls test (*p* < 0.05). Vertical bars represent the standard error of the means.

**Figure 2 plants-08-00007-f002:**
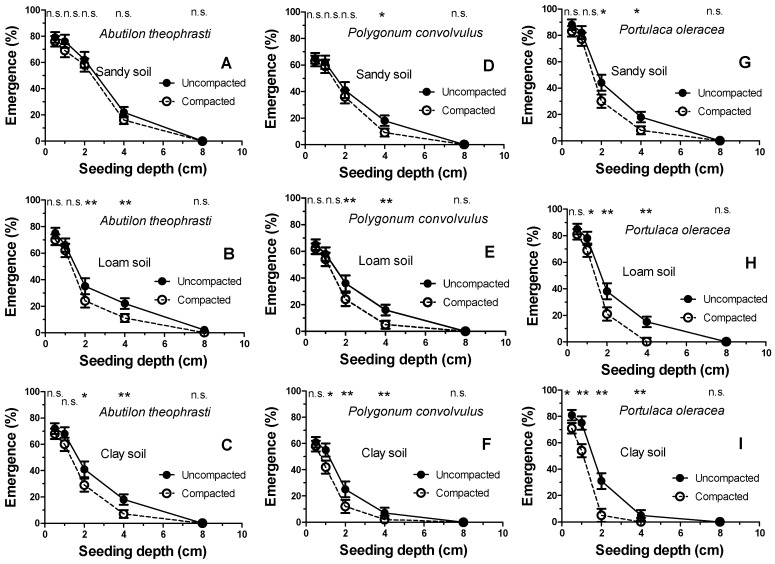
Seedling emergence of the three selected weeds as a function of sowing depth (0.5, 1, 2, 4, and 8 cm), texture (sandy (**A**,**D**,**G**), loam (**B**,**E**,**H**), and clay soil (**C**,**F**,**I**)) and soil compaction. Single or double asterisks indicate the significance level of pairs of data (each soil depth) in the Student–Newman–Keuls test (*p* < 0.05 or *p* < 0.011, respectively).

**Figure 3 plants-08-00007-f003:**
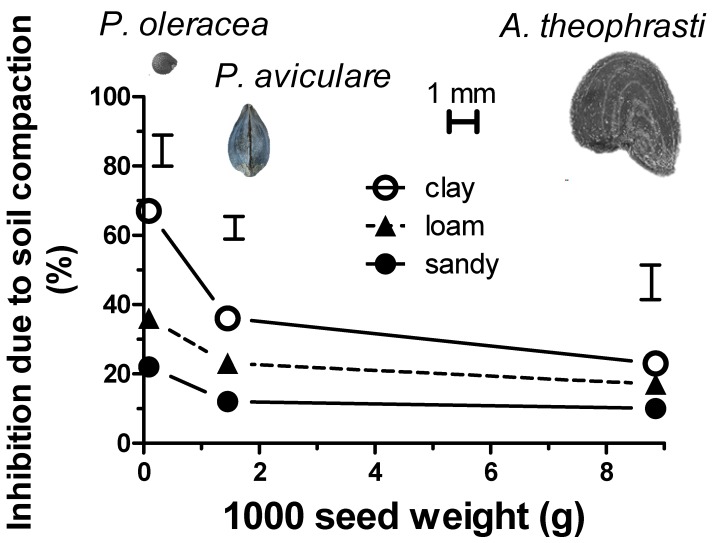
Emergence inhibition due to soil compaction (difference between tilled and compacted soils of the three different soil textures) as a function of the 1000 seed weight of the selected weed species. For all species, the inhibition was calculated at a sowing depth of 2 cm. The horizontal bar indicates 1 mm in order to provide reference for the real seed size. Vertical bars indicate the LSD (Least Significant Difference) values in the Student–Newman–Keuls test (*p* < 0.05).

**Figure 4 plants-08-00007-f004:**
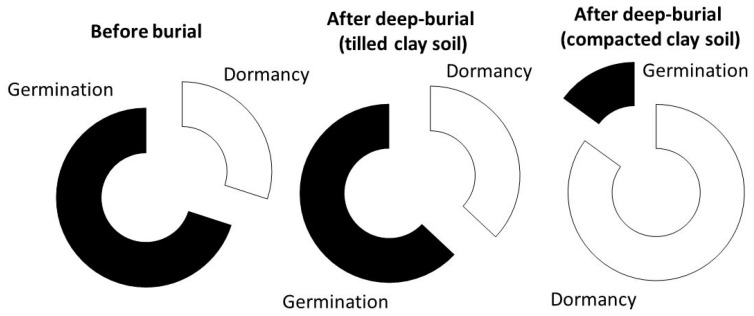
Germination and dormancy percentage of *Portulaca oleracea* seeds before and after deep-burial in tilled and compacted clay soil.

**Table 1 plants-08-00007-t001:** Chemical and physical characteristics of the soils used for the experiments.

Soil Types	Sand (%)	Clay (%)	Lime (%)	CaCO_3_ (%)	pH	Organic Matter (%)
Sand	93	2	5	5.1	7.3	0.3
Loam	65	16	19	4.2	7.1	1.4
Clay	30	35	25	5.5	7.5	1.8

**Table 2 plants-08-00007-t002:** Seed weights of the three studied weeds.

Species	Botanic Family	1000 Seed Weight (g)
*Abutilon theophrasti* Medicus	Malvaceae	8.850 ± 0.87
*Polygonum convolvulus* L.	Polygonaceae	1.455 ± 0.28
*Portulaca oleracea* L.	Portulacaceae	0.094 ± 0.09

**Table 3 plants-08-00007-t003:** Exhumed seed germination and dormancy performances of the three weed species as a function of soil texture and compaction during prolonged burial (six months at 8 cm depth). Single or double asterisks indicate, for each data pair, significant differences in the means (for *p* < 0.05 and *p* < 0.01, respectively, ns = not significant) in the Student Neumann Keuls (SNK) test.

Species	Soil Ecology		Seed Performance after Prolonged Burial							
	Texture	Compaction	Germination (%)		Light Dormancy (%)		Deep Dormancy (%)		Dead Seeds (%)	
*Abutilon theophrasti*	Sand	Tilled	72.2	ns	9.8	ns	15.5	ns	2.5	ns
		Compacted	68.5		9.5		19.7		2.3	
	Loam	Tilled	55.0	ns	8.6	ns	34.4	ns	2.0	ns
		Compacted	47.8		4.2		44.9		3.1	
	Clay	Tilled	53.7	**	7.3	ns	36.5	**	2.5	ns
		Compacted	24.2		4.8		68.5		2.5	
*Polygonum convolvulus*	Sand	Tilled	68,2	ns	11.8	ns	17.6	ns	2.4	ns
		Compacted	65.1		9.9		21.8		2.8	
	Loam	Tilled	52.5	*	6.5	ns	38.8	*	2.2	ns
		Compacted	33.0		5.5		56.5		3.0	
	Clay	Tilled	48.4	**	8.6	ns	40.5	**	2.5	ns
		Compacted	21.6		4.4		71.6		2.4	
*Portulaca oleracea*	Sand	Tilled	77.5	ns	8.5	ns	11.2	ns	2.8	ns
		Compacted	75.2		4.8		17.5		2.5	
	Loam	Tilled	68.9	**	5.1	ns	22.8	**	3.2	ns
		Compacted	41.5		2.5		53.6		2.4	
	Clay	Tilled	37.0	**	4.5	ns	56.4	**	2.1	ns
		Compacted	13.4		1.6		83.3		1.7	

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
