# Peer review of "Soil Physics Involvement in the Germination Ecology of Buried Weed Seeds"

_plants, 2018, doi:10.3390/plants8010007_

Round 1

Reviewer 1 Report

The paper is original and well written, clear and easy to read. Data and results are presented appropriately.  

I have only two minor comments to the Figures.

Figure 1. should be improved. The bars in a good way show the differences in the bulk density before and after compaction, but the scale is misleading. Clay soil should be characterized by the highest bulk density (am I wrong?)

Figure 3. Dot is missing in the name of A theophrasti

Author Response

Dear referee I replay to your kind suggestion points:

1) Indeed, when we talk about soil properties, I risk getting the ladder wrong. However I checked carefully and I verified that the clayey soil has a lower bulk density due to the higher porosity (empty spaces occupied by the air)

2) In the figure 3 the lines (dotted or continue) indicate the dimensions of the soil particles. At the top of the graph were shown the seeds, with their names, only to indicate their shape and size. in any case there are all three names of weeds

Reviewer 2 Report

The researches done and the results are biologically and technologically relevant. The issues related to the weed seeds dormancy mechanism of which the longevity of soil seedbank and its importance for the farm hygiene depends are always welcome for weed sciences.

The studied weed species are also important for many crops from European counties and beyond, and the different soil texture from the experiment and the different compaction rates are very important factors from an ecological and agronomical point of view.

Methods are appropriate described, results comprehensive and statistically correct, the discussions are supported by other relevant authors results, the conclusions well defined by the own results and the appropriate literature.

Please verify the figure 1(bulk density)! There are some errors.

Author Response

Dear referee 2 replay to your kind suggestion points:

I verified that the clayey soil has a lower bulk density due to the higher porosity (empty spaces occupied by the air).

Reviewer 3 Report

This ms deals with relationships of germination success of typical weed species and the soil physical environment (depth, texture, compaction). The topic is of high relevance with respect to agricultural practice (weed seed banks) as well as an understanding of the reproduction ecology of plants. The author carried out a series of two connected experiments and found that burial depth, soil compaction, soil texture and seed size influenced germination success. A combination of deeper burial, clay texture, small seed size and soil compaction led the strongest inhibition of germination.

The study is well-done and the ms is well-prepared. There are not many objections. 

With respect to statistics I wonder why you do not present the main and interaction effects of the ANOVA. Especially interactions are of interest to work out dependencies of effects of single tested factors, and, obviously, such dependencies are given here. I think, a presentation of main and interaction terms would essentially enrich your ms.

The Results section should be shortened by at least 10-20 %. There are some redundant parts (see Specific remarks) which could be deleted.

Overall, this is a valuable study which enriches existing research on germination in agricultural soils.

Specific remarks (page/line):

1/10: I would prefer to read a short comment on the background of your study in the Abstract (weed seed banks, agricultural practice).

1/19: Add time period of seed storage in the soil.

4/21: How long were the seeds stored? Please, add this information in your ms.

4/26: The experiment was conducted in spring 2016, but the seeds used in this experiment were collected in summer 2016. Please, check the dates.

4/28: mean daily temperature?

5/1-4: This sentence should be rephrased. Suggestion: “Three agricultural soils were chosen according to different textures: sand, loam, clay (Table 1).” Please, unify terminology through your ms (sandy, sand).

5/6: Put a comma: […] existing seed bank, soils were obtained […]

5/6: Used soil substrates were gathered from depth of >0.4 m. Is this depth below the ploughpan? Please comment on that point in your ms, because it is relevant for the seed density in the gathered soil. Have you established a seed bank control by means of pipes without sown seeds? What is about seed rain? Was aerial seed input controlled?

5/7: […] and the relative soil […]

5/13-24: Could you make more clearly that half of your pipes were compacted and the other half not? I had particularly problems in understanding the expression “The same, above cited, soil types were compacted […]”.

5/14: Add specification to make sentence more clearly: Two hundred seeds [...] were sown at each of the following depths: 0.5, 1, […]

5/27-28: How often have you counted the seedlings?

6/3-7: This sentence is very long and hard to understand. Particularly, I do not understand the term “compaction procedure” in this context.

8/8: […] for P. aviculare and P. oleracea a strong […]

8/11: Looking at Fig. 2, I wonder whether it was a complete or almost complete inhibition in case of Polygonum.

8/17-18: These three values are redundant to the information given in Table 2. I suggest to delete them in the text and only comment on the fact of a huge difference.

8/19: Better place “(Figure 3)” behind “theophrasti”

8/22-25: Was this value (70 %) calculated across soil depths? Please, clarify.

8/25: Insert parentheses to make context more clearly: “Conversely, the heaviest seeds (A. theophrasti) showed […]”

9/11-13: redundant information, could be deleted

9/26: delete the word “of”

10/15: Why at intermediate depths? Is there an explanation for this phenomenon?

Figure 1: uniform terminology (sand/loam/clay).

Figure 3: Add a scale or a comment on seed sizes.

Table 1: How where these soil variables measured? How many samples were analyzed per value? Why was there no linear correlation between lime and CaCO3 values?

Author Response

1/10: I would prefer to read a short comment on the background of your study in the Abstract (weed seed banks, agricultural practice).

Ok, I agree. In the second line I have inserted this sentence:

Indeed, these ecological conditions, due to the adopted agronomic practices, play a crucial role in modulating the seed bank germination dynamics.

in fact, these ecological conditions play a crucial role in modulating the germination dynamics of the seed bank according to the different agronomic practices adopted

1/19: Add time period of seed storage in the soil.

Ok, I have inserted in the abstract “for six months”.

4/21: How long were the seeds stored? Please, add this information in your ms.

Ok, I specified this in the text:

…sown (after about eight months after collection)…

4/26: The experiment was conducted in spring 2016, but the seeds used in this experiment were collected in summer 2016. Please, check the dates.

Ok you are right, I entered the right date (pag.3 line 27) of the collection occurred in summer 2015.

4/28: mean daily temperature?

It was already reported in; Pag. 4 line 7 (mean temperatures ranged (min-max) around 15/25°C).

5/1-4: This sentence should be rephrased. Suggestion: “Three agricultural soils were chosen according to different textures: sand, loam, clay (Table 1).” Please, unify terminology through your ms (sandy, sand).

Ok, I have inserted your kind suggestion. In addition I have uniformed the term used “sandy” instead of “sand”. Even in figure 1 and figure 2.

5/6: Put a comma: […] existing seed bank, soils were obtained […]

Ok, It was made

5/6: Used soil substrates were gathered from depth of >0.4 m. Is this depth below the ploughpan?

Please comment on that point in your ms, because it is relevant for the seed density in the gathered soil.

Ok, I have inserted (pag.4 line 13): below the ploughpan…

Have you established a seed bank control by means of pipes without sown seeds? What is about seed rain? Was aerial seed input controlled?

Ok, I have inserted these opportune specification in the text:  pag 4 (line 30-31) and 5 (line 1-2):

Additional polypropylene pipes, with the only non-sown soil, confirmed (data not shown) the virtual absence of any seed banks. In addition the area surrounding the polypropylene pipes was kept free of weeds to avoid any unwanted “seed rain” inside the cylinders.

5/7: […] and the relative soil […]

OK

5/13-24: Could you make more clearly that half of your pipes were compacted and the other half not?

Ok I specified so (pag. 5 line 17-18): Emergence test were carried out in half pipes with compacted soil and the other half not (tilled soil).

I had particularly problems in understanding the expression “The same, above cited, soil types were compacted […]”.

Ok, you are right. Now I have specified that the additional sowing was made at 8 cm depth. (pag. 5 line 24)

5/14: Add specification to make sentence more clearly: Two hundred seeds [...] were sown at each of the following depths: 0.5, 1, […]

ok, I've simplified this sentence (pad 4 line 19-20).

5/27-28: How often have you counted the seedlings?

Ok. I specified at the cotyledon appearance.

6/3-7: This sentence is very long and hard to understand. Particularly, I do not understand the term “compaction procedure” in this context.

OK, I agree. Now I have simplified this sentence (pag. 5, line 5-10).

8/8: […] for P. aviculare and P. oleracea a strong […]

OK, it was corrected

8/11: Looking at Fig. 2, I wonder whether it was a complete or almost complete inhibition in case of Polygonum.

It was an almost complete inhibition in the case of clay soil.

8/17-18: These three values are redundant to the information given in Table 2. I suggest to delete them in the text and only comment on the fact of a huge difference.

Ok, this redondance was deleted in the text.

8/19: Better place “(Figure 3)” behind “theophrasti”

It is difficult to move because the sentence starts with "figure 4". I'd rather leave it like this

8/22-25: Was this value (70 %) calculated across soil depths? Please, clarify.

Ok, I clarified “to respect to tilled soil” Pag.7 line 21)

8/25: Insert parentheses to make context more clearly: “Conversely, the heaviest seeds (A. theophrasti) showed […]”

OK, it was insterted between parenthesis

9/11-13: redundant information, could be deleted

Ok, I deleted this sentence

9/26: delete the word “of”

Ok, sorry… it was deleted

10/15: Why at intermediate depths? Is there an explanation for this phenomenon?

This is more evident at intermediate depths because at lower depths the inhibition is, often, too slight while at higher depths it is too inhibitory for all (example 8 cm inhibition for all 100%)

Figure 1: uniform terminology (sand/loam/clay).

Ok, it was made

Figure 3: Add a scale or a comment on seed sizes.

.OK I have inserted a bar of 1 mm in the graph and in the text I have inserted “Horizontal bar indicate 1 mm for to indicate the real seed size”.

Table 1: How where these soil variables measured? How many samples were analyzed per value? Why was there no linear correlation between lime and CaCO3 values?

These soils had already been used previously and analyzed according to international standards. They were also chosen to have a similar pH so as not to interfere chemically in order to evaluate the only soil physics in respect to the germination of buried seeds.